# Teaching and Rehabilitation of Handwriting for Children in the Digital Age: Issues and Challenges

**DOI:** 10.3390/children10071096

**Published:** 2023-06-22

**Authors:** Nathalie Bonneton-Botté, Ludovic Miramand, Rodolphe Bailly, Christelle Pons

**Affiliations:** 1Laboratoire de Psychologie: Cognition, Comportement, Communication (LP3C), University Brest (UBO), 29000 Brest, France; nbonneto@univ-brest.fr; 2Pediatric Rehabilitation Department, Fondation Ildys, Rue Alain Colas, 29200 Brest, France; ludovic.miramand@ildys.org (L.M.); rodolphe.bailly@ildys.org (R.B.); 3LaTIM (Laboratory of Medical Information Processing), INSERM UMR 1101 (Institut National de la Santé et de la Recherche Médicale, Unité Mixte de Recherche), 22 Avenue Camille Desmoulins, 29238 Brest, France; 4Department of Physical Medicine and Rehabilitation—Brest University Hospital Center, 2 Avenue Foch, 29200 Brest, France; 5UFR (Unité de Formation et de Recherche) Médecine, University Brest (UBO), 22 Avenue Camille Desmoulins, 29238 Brest, France

**Keywords:** handwriting, graphomotor learning, learning disabilities, digital technology, artificial intelligence, inclusive society

## Abstract

Handwriting is a determining factor for academic success and autonomy for all children. Making knowledge accessible to all is a challenge in the context of inclusive education. Given the neurodevelopmental diversity within a classroom of children, ensuring that the handwriting of all pupils progresses is very demanding for education professionals. The development of tools that can take into account the variability of the profiles and learning abilities of children with handwriting difficulties offers a new potential for the development of specific and adapted remediation strategies. This narrative review aims to present and discuss the challenges of handwriting learning and the opportunities offered by new technologies involving AI for school and health professionals to successfully improve the handwriting skills of all children.

## 1. Introduction

Handwriting is a complex activity that involves an array of component skills and continuous interaction between lower-level motor skills, higher-order cognitive processes, and the environment [1,2,3,4]. Around 30% of typically developing children have difficulty learning handwriting; for around 10% of children, these difficulties are considerable and long-lasting with repercussions in all situations of daily life [5,6,7]. The proportion of children with handwriting difficulties is even greater among those with neurodevelopmental and/or motor and/or visual disorders [8,9,10,11].

Writing plays a key role in a range of learning activities, such as reading, spelling, and mathematics [12,13,14,15,16]; thus, children with handwriting difficulties have an increased risk of schooling problems. Moreover, teachers attribute lower marks to poorly legible work [17]. Therefore, handwriting difficulties lead to negative learning experiences, which may result in low self-esteem [18,19]. Considering the international will to develop the accessibility of education from an inclusive perspective, handwriting difficulties have a global and negative impact on schooling. Improving strategies to identify, prevent, and remediate those difficulties is a major challenge needing specific attention from school teachers and rehabilitation professionals involved in the acquisition of handwriting skills. 

New opportunities are being created by unprecedented technological advances. Digital learning technologies including artificial intelligence (AI) can be used both as educational tools and as assistive technologies that facilitate access to education and/or learning materials, allow participation in learning activities and/or overcome barriers to learning [20]. Global organizations such as the WHO and UNICEF perceive such technologies as a means to improve the equality of education and health management, facilitate inclusion, and provide open and personalized interventions [20,21,22,23]. In the context of handwriting learning, those technologies may provide new tools for practice, assessment, and remediation. In this review, we consider remediation as spanning both the education and rehabilitation fields. Artificial intelligence (AI) may lead us to rethink handwriting learning and remediation. Devices that use AI are already being introduced into school classrooms and rehabilitation [24,25], which raises the question of their place in these contexts. 

This narrative review aims to present and discuss the challenges of handwriting learning and the opportunities offered by new technologies involving AI for school and health professionals to successfully improve the handwriting skills of all children. We first describe the specificities of handwriting learning and the challenges relating to handwriting acquisition, evaluation, instruction, and remediation. Then, we present the potential of technologies involving AI for handwriting learning and discuss their perspectives as well as the challenges relating to their implementation in education and rehabilitation practices. 

## 2. Methods

This narrative review was conducted according to the Scale for the Assessment of Narrative Review Articles (SANRA) criteria [26]. It was specifically designed to consider and synthesize data from the educational and rehabilitation sciences regarding handwriting learning and the opportunities offered by new technologies for the acquisition of this skill for all children. 

A literature search was performed between July 2022 and April 2023 by the 4 authors using the following terms: handwriting; dysgraphia; motricity; diagnosis; evaluation; analysis; intervention; rehabilitation; therapy; education; collaboration; technology; artificial intelligence. Relevant scientific studies were identified in Medline (Pubmed), Web of Science, Google Scholar IEEE, and Scopus. A comprehensive search of the references of selected papers was conducted to identify further articles of interest. Special attention was paid to systematic reviews and meta-analyses. Articles published in French and English in peer-reviewed journals were considered. We considered studies of new technologies published from 2010 to provide up-to-date data and trends in this field. We paid specific attention to devices that have already been introduced in teaching and rehabilitation programs. 

We synthesized the evidence with regard to the goals of this narrative review. The authors met regularly to cross-check ideas and expertise to (i) identify and select key statements and (ii) organize the presentation of the data.

## 3. Handwriting Acquisition Is a Complex Learning Process

Handwriting has been described as a “complex perceptual-motor skill encompassing a blend of visual–motor coordination abilities, motor planning, tactile and kinesthetic sensitivities but also cognitive, and perceptual skills” [19]. It is also a prerequisite for higher-order processes required later in life, such as literacy.

### 3.1. Contribution of Models to the Understanding of Learning and Motor Control

Although no specific developmental models have been defined for handwriting learning, the process of handwriting learning can be understood through the major models of learning and motor control. These models can be used to formulate different explanatory hypotheses when the acquisition and execution of handwriting is problematic. 

Motor learning is a process that aims to improve the performance of a specific motor skill. The motor schema theory [27], a reference in the rehabilitation field, considers the existence of a recall schema whose function is to retrieve from memory a generalized motor program to initiate the movement to be produced and a recognition schema that integrates the sensory consequences of the movement (based on past experiences). The interaction between these two schemas and knowledge of the result allows the progressive acquisition of motor skills. Therefore, this model is based on the ability to memorize the relationships between the initial conditions of a motor action and the sensory results of that action as well as knowledge of performance (knowledge of the result and/or process). Research underpinning this model has highlighted the importance of practice, experience, and variability in the learning context, as well as feedback to improve recall and recognition schemas [28]. Feedback refers to knowledge of the body’s action and its outcome (knowledge of the result and process) that can be broadly subdivided into intrinsic (inherent feedback) and extrinsic feedback (or augmented feedback) [29]. Intrinsic feedback refers to the processing of available sensory information (e.g., hand movement and vision of the handwriting product); whereas extrinsic feedback involves additional information from an external agent (e.g., a teacher or a tablet) or supplementary sensory information (e.g., sound) that may or may not be processed by the individual.

Improving motor skill performance requires improving motor control by motor learning. Motor control can be modeled using a computational and probabilistic theory of internal models of action [30,31,32]. This theory distinguishes two internal and complementary models of action: the inverse model and the direct model. The inverse model generates the appropriate motor commands to achieve the desired action based on the initial condition. The direct model also transmits an efference copy to predict the sensory consequences of the upcoming movement, called forward model. This prediction of sensory consequences feeds into a comparator, which also receives sensory feedback from the ongoing movement. The comparison between actual and predicted sensory feedback is used to detect whether an “error” has occurred during movement execution. If necessary, corrections are made to adjust the motor commands [33]. The acquisition of skill thus involves learning to map between motor commands and sensory consequences. The ability to extract and memorize information from the environment during learning, the importance of practice, and the nature of feedback are emphasized by the theory of internal models of action.

In summary, according to these two complementary models, motor learning and motor control are possible because of (i) the sensory feedback produced during motor action, and (ii) the knowledge that the learner obtains about the characteristics of their movement in relation to their initial intention (i.e., knowledge of the result and/or the process) [27,30]. 

Although these two models are references in the field of health and education research, they are incomplete. They are mainly based on empirical results obtained from adult learners, most often during much simpler tasks than handwriting learning. This specific task involves a remarkable learning process insofar as it takes place over several years, during a period in which neuro-motor maturation is not completed. The ability to integrate sensory feedback, to represent internal models of movement, and to articulate motor and linguistic learning (i.e., to understand the link between the learned gesture and a language unit) is not trivial [34]. Our literature search did not yield articles about the non-linear dynamics of motor learning theory; however, this theory might be considered as a complementary approach [35]. Moreover, handwriting, by its linguistic dimension, is a specific motor learning process whose acquisition takes place in different learning conditions with different agents (e.g., different teachers, parents, and rehabilitation professionals) who can provide a variety of extrinsic feedback, but that could potentially be contradictory. 

### 3.2. Developmental Benchmarks for the Acquisition of Handwriting

The developmental progression in the control of the graphomotor gesture, which occurs during childhood and adolescence, completes the motor learning models described above. It shows that the progressive automation of handwriting is related to the development of the ability to access intrinsic feedback to construct, memorize and use internal models of action, and to accurately take into account the knowledge of results [36,37]. The ability to integrate intrinsic feedback is age dependent. Before the age of 5–6 years, the child is not capable of integrating visual and proprioceptive information to correct movement during its execution [37]. Thus, the strokes that make up the first writing attempts are performed using ballistic movements. Between the ages of 6 and 8 years, sensory feedback processing plays an important role in the execution of handwriting. Dependence on sensory information requires sustained attention, thus the attentional resources of young writers are mainly allocated to controlling the handwriting rather than to other aspects of writing such as spelling and content [38]. During the early stages of motor learning, handwriting control is essentially based on the use of visual information and little on proprioceptive information [39]. Visual and attentional skills play a major role at this stage. The ability to integrate visual information is particularly important in the early years of handwriting learning to correct the action and to elaborate motor programs [40,41]. Proprioceptive function does not appear to be fully mature until 7 years of age [42], and at this age proprioceptive information cannot be used as a reference for movement execution [39]. Between the ages of 10 and 14 years, a balance is reached between the two modes of feedback and feedforward control: writing becomes automated, fluid, and fast, sometimes to the detriment of legibility [43,44].

International studies have reported that between 6% and 30% of children have difficulty learning to write. Several longitudinal studies have demonstrated a large variability in acquisition time [5,6,45]; some children will achieve in one year what others will require 3 years to acquire. This could be explained by cultural and environmental factors as well as the child’s level of cognitive development [46]. From a cognitive point of view, conceptual knowledge of written language could go a long way towards explaining inter-individual variability. Children between the ages of 3 and 5 years acquire both universal and specific knowledge However, according to Puranik and Lonigan (2011), children first need to acquire universal knowledge about the written language system (i.e., when they pretend to write, young children produce linear, segmented strokes composed of simple shapes) and then specific knowledge linked to the linguistic system (i.e., directionality, symbol shape associated with the name and sound of letters) [4]. Between the ages of 5 and 7 years, children develop the ability to write letters from memory and fluently [47,48], which is essential for orthographic and compositional skills. On the graphomotor level, writing consists of transcribing the letters that make up words, and a delay in the conceptual knowledge of linguistic units might contribute to explain the variability observed on the gestural level. These developmental sequences can guide understanding of how gaps are created between pupils during the first years of school. Links between procedural and conceptual knowledge remain largely under-explored.

### 3.3. Handwriting Disorders in Children

In the scientific field of handwriting, terminologies are polysemous and plural, and their acceptance may differ from one country to another [49]. We will use the term disorder for any situation in which the child’s handwriting constitutes a hindrance to his or her schooling, whether this hindrance is transient or permanent. Handwriting disorders are frequent and of multiple forms. Writing may be slow and degraded or both. Some children may complain of tiredness and muscle pain but not all. Different hypotheses have been proposed to explain these disorders. According to Wann and Jones (1986), impairment of the temporal organization (i.e., dysfluency and high pause time) of writing reflects a defect in motor programming because of an over-reliance on visual feedback [50]. Zesiger (2003) and Van Galen et al. (1993) suggested that difficulty inhibiting neuromotor noise (i.e., over-representation of high-frequency movement) affects the motor execution component of the task, generating spatial, temporal, and kinematic irregularities [51,52]. Lopez and Vaivre-Douret (2021) proposed a multi-factorial hypothesis to explain difficulties with movement representation, associated with poor motor control [49]. They suggested that impaired visual–kinesthetic integration might be the cause of control difficulties for many young writers. 

Handwriting disorders are particularly frequent in children with neurodevelopmental and/or motor and/or visual disorders [8,9,10,11], highlighting the multiple causes of these disorders. Children with cerebral palsy frequently have poor quality handwriting because of difficulty maintaining neatness at speed and over long periods of time. This has been linked to impairment of proprioception, tactile sensory, visual and spatial perception, and visual–motor coordination, as well as of dexterity, fingertip-force, coordination, movement planning skills, and postural control [53,54,55]. The development of handwriting and of fine motor skills is longer in children with cerebral palsy than typically developing children [54]. Children with attentional deficits with or without hyperactivity have handwriting disorders involving spelling errors and poor legibility [56]. These children require more time to write, especially for long words, which they correct excessively. The production of letters and words is inconsistent, and letters are disproportionate in size. These children increase the pressure on the pen to improve control during writing, but this high pressure does not allow them to increase fluency and might be a cause of hand pain during handwriting. Spelling difficulties may not be related to a defect in spelling representations but rather to a deficit in the retention of information at the level of the graphemic buffer [56]. The poor writing performance might result from attentional and motor difficulties rather than linguistic difficulties [56].

### 3.4. The Measurement Problem

The evaluation of handwriting involves first assessing the writing itself and then assessing any associated disorders to deepen understanding of the impairment and guide remediation.

Numerous handwriting evaluation scales have been developed over the last half-century [57,58], based on teacher and therapist experience (i.e., construct validity). Some are questionnaires intended for the evaluators [59] whereas others target the child [60]. For example, the Handwriting Proficiency Screening Questionnaire (HPSQ) assesses three domains: legibility, performance time, and physical and emotional well-being. Other tools are based on tests. The tests commonly used in practice and in research differ from one country to another. Some examples are the Concise Evaluation Scale for Children’s Handwriting or *Brave Handwriting Kinder* (BHK) [61,62,63], the Minnesota Handwriting Test [64,65], Evaluation Tool of Children’s Handwriting (ETCH) [66], the Hebrew Handwriting Evaluation (HHE) [67], the Detailed Assessment of Speed of Handwriting (DASH) [68], and the Handwriting Legibility Scale (HLS) [69]. 

These tests mainly focus on the letters written, writing speed, and sometimes the ergonomic features of writing (e.g., pressure). Handwriting speed is relatively easy to determine (i.e., letters per minute); however, evaluation of the letters written and the ergonomic features of the gesture remain subjective. Therefore, detailed criteria have been proposed to quantify these features. For letter writing, the most common evaluation criteria are the size, shape, space, tilt, and straightness to the line [19,57,58]. Fewer ergonomic criteria have been defined and they essentially relate to pencil grasp and pressure, and paper stabilization [66,67]. 

Since many criteria are dependent on the factors that influence the performance of this complex sensorimotor task [19,70], most handwriting evaluations try to standardize the conditions (i.e., task duration and text copying), the content (i.e., text, words, numbers, and forms with variations according to the written language), the instruction (write as usual or as well as or as quickly as), and the accessories used (paper and pen).

The psychometric properties of those tests have been investigated in a few studies mostly conducted by the scale developers [71]. Intra-rater reliability is moderate to excellent [72] depending on the test [19,57] and the evaluator’s experience [64,65]. Inter-rater and test-retest reliability [64,65,68,73,74] have been less investigated and only in a few languages (i.e., English, Chinese, and Persian), limiting transferability between professionals as well as longitudinal follow-up assessments [75]. Handwriting evaluations are specific to each alphabet [76] and to the child’s school grade [70], and cross-cultural differences have been identified [77,78,79]; therefore, the translation of each assessment tool should be associated with an update of the normative data for comparison of the child’s handwriting [71]. 

A limitation of handwriting evaluations is that they do not identify the cause of the difficulties. Therefore, some assessments, which are mainly performed by rehabilitation professionals, assess associated difficulties such as the fine motor coordination, visual–motor integration, and kinesthesia of handwriting [80]. Among the commonly used tests in research and clinical practice [81], the Beery Developmental Test for Visual Motor Integration (Beery VMI) and the Nine Hole Peg Test (9HPT) (which, respectively, assess visual motor integration and fine motor coordination) are the only tests that can be performed without a pen. The Bruininks–Oseretsky Test of Motor Proficiency, second edition (BOT-2) and the Denver Developmental Screening Tests–second edition (Denver II) assess both visual motor integration and fine motor coordination, whereas the Movement Assessment Battery for Children, second edition (M-ABC-2) and the School Version of the Assessment of Motor and Process Skills (School-AMPS) specifically assess fine motor coordination. The psychometric properties of these tools have been evaluated to a greater extent than those of the handwriting assessment tools [81,82]. However, the cross-cultural validity and the test-retest reliability of these tests remain undervalued [82] despite their usefulness in research and practice [81]. 

Handwriting assessments are affected by the subjectivity of the evaluator and psychometric properties of the tool; therefore, conclusions regarding their quality are difficult to draw, especially for the longitudinal follow-up of a child’s handwriting. Moreover, the assessment focuses on the writing itself and does not consider the gesture or the manipulation of the pen. Clinical assessments of fine motor coordination, visual motor integration, and kinesthesia partially complement handwriting assessments by providing information regarding the impairments that cause the handwriting problem.

### 3.5. Considerations

Despite the existence of motor learning theories and developmental benchmarks of handwriting acquisition, no integrated model of handwriting acquisition that takes into account the child’s motor and cognitive development and, especially, their relationship to the linguistic system of their mother tongue currently exists. The large variety of handwriting difficulties further complicates their understanding. Furthermore, the assessments used to evaluate handwriting difficulties around the world are heterogeneous, and data are not comparable across countries. Therefore, there is currently a lack of knowledge and understanding of inter-individual variability and the limits between typical development and disorders that might impact other learning domains.

## 4. Handwriting Instruction: A Key Factor in Learning and Rehabilitation

### 4.1. General Principles for Learning for All

Handwriting is a biologically secondary ability whose acquisition is strongly determined by culture and modes of transmission [83]. This specific knowledge domain requires structured and formal instruction as well as the motivation to learn. A growing body of experimental evidence-based studies has led to the identification of some broad principles for handwriting learning for all children.

Practice is key for the successful learning [28] and improvement [1,2,3,19,49,84] of handwriting. To be beneficial, an intervention for handwriting difficulties should be performed at least twice weekly for 10 weeks [84].Approaches should be task-specific [1,2,3,19,49,84]. A task-specific approach improves handwriting legibility. It is reasonable to believe that improvements in letter quality may precede improvements in speed; speed likely requires additional practice time although this remains to be demonstrated [84]. Sensorimotor interventions that address isolated component skills, such as visual perception, kinesthesia, in-hand manipulation, visual–motor integration, or biomechanical features of handwriting have no effect on handwriting legibility and should not be used [1,2,3,19,49,84].Handwriting instruction should be explicit. Legibility and fluency are more likely to improve with an explicit handwriting program than no instruction or non-handwriting instructional conditions [85].Intrinsic and extrinsic feedback of sufficient quantity and quality should be available. Feedback is a crucial parameter in the learning process. A recent review of extrinsic augmented feedback to improve handwriting performance and learning showed that the effectiveness of the feedback varies according to its timing (immediate feedback is more beneficial than delayed feedback) and the nature of the information provided to the learner (rich, precise feedback has a greater impact on learning). Feedback on performance can be accepted, modified, or rejected by the individual [86] and thus may have a positive or negative effect on learning [87].Variability should be introduced during handwriting learning. Varying the letter font and size during learning facilitates memorization [88].Learning environments should be motivating, supportive, and include self-regulation of performance [89].

Those principles are in accordance with the principles of motor learning theories and the principles of non-linear pedagogy [90] and emphasize the relevance of providing personalized handwriting instruction.

When a child has difficulty with learning handwriting, remediation may be proposed. Remediation is “a class or activity (including rehabilitation) intended to meet the needs of students who initially do not have the skills, experience, or orientation necessary to perform at a level that the institution or instructors recognize as “regular” for those students” ([91], p. 174). 

### 4.2. The Contribution of Rehabilitation Principles

Rehabilitation is a problem-solving process that aims to equip children and adults to live autonomously, fulfill their maximum potential, and optimize their contribution to family life, their community, and society. It is a philosophy of care based on the biopsychosocial model of illness that helps to ensure inclusion within the community, in employment, and in education. Rehabilitation may benefit any child with a long-term condition [92,93]. 

Children identified by their teachers as having poor handwriting legibility and who do not respond to classroom interventions are frequently referred for rehabilitation, particularly occupational therapy [84]. Rehabilitation is based on individualized assessment and remediation. It is effective if the general principles detailed above are respected, and various types of approaches exist [2,3,49,84]. 

Due to the considerable variability in the presentations and abilities of children, a specific evaluation of both the handwriting and associated difficulties must be undertaken to gain a better understanding of each child’s individual problems [19,94]. The evaluation should also consider all factors that could potentially affect the child’s handwriting ability at home and school, such as their family, friends, and teachers, their physical environment, as well as personal factors such as motivation and coping strategies [92,94]. This will facilitate the proposal of an individualized remediation strategy.

The planning of an individualized intervention should begin by understanding the child’s goals to ensure consideration of the personal factors of motivation and interests, as defined by the International Classification of Functioning (ICF) [94,95]. The child must first be invited to identify the skills and abilities that are most important to them, then small, realistic goals should be set with them and their parents to improve motivation and outcomes [94,95]. Once the goals have been determined, professionals should carry out structured observations and task analysis of the child attempting their goal to determine the factors that limit goal achievement [92,95]. 

Several curriculum-based handwriting programs in which teachers and occupational therapists collaborate [96] have shown promising results on handwriting legibility [96]. These programs are taught within the classroom setting and are geared toward improving handwriting in all children, not only those with difficulties. 

As in all rehabilitation interventions, both the child and their family must be supported [92,97,98]. Professionals should involve and empower the parents to enable them to participate in supporting the child, to continue interventions at home as a supplement to therapy sessions, and to cooperate with professionals [94,99]. Collaboration between rehabilitation professionals, the children, their parents, and teachers is essential. Handwriting practice should be organized in a coordinated way between home, school, and rehabilitation [1,84,96]. Good coordination ensures the continuity of the learning method, as well as a sufficient amount of training.

### 4.3. Emerging Principles for Learning and Rehabilitation

Many techniques exist for handwriting learning and remediation. Recent studies have highlighted the effectiveness of specific techniques that could guide the development of future methods. 

Studies in young, typically developing children have shown that the use of gross motor skills (i.e., writing in the air and walking on letters positioned on the floor) improves the construction of the motor representations of letters [100]. 

For children in elementary school, cognitive strategies such as self-evaluation techniques (children receive instruction to help them think their way through letter formation and to self-correct) can be useful for children with handwriting difficulties [101,102]. Motor imagery, a technique used in rehabilitation and sports training that consists of imagining oneself performing a movement, also appears to improve handwriting skills [103]. Letter production improved in elementary school children with developmental coordination disorders after they watched video sequences showing how to form letters, performed mental simulation exercises of the writing movement, and alternately made real written productions [104]. The improvements were greatest in those who achieved an adequate level of motor imagery.

Regarding feedback provision, trace deletion during letter tracing might improve fluency and speed abilities for typically developing children [100] and adults [105].

Movement sonification, i.e., the transformation of movement into sound provided as a feedback, enables the writer to access kinematic information, which appears to facilitate the development of motor programs, especially for children with slow and disfluent writing [106].

However, evidence to support those interventions remains scarce. Studies have only been performed in small groups of children. Further research is needed to improve the level of evidence and to determine the types of interventions that are the most effective in different situations, e.g., typically developing children, children with handwriting difficulties with no specific diagnosis, children with specific impairments or pathology, etc. Studies are also needed to determine the optimal timing of remediation according to the stage of development, the quantity of training needed, etc. [3,84]. 

### 4.4. Learning and Rehabilitation in Practice

Several recent studies, mostly surveys and observational studies, have identified that commonly used remediation practices in school are not always based on the general learning principles identified in Section 4.1. [107,108,109,110]. They also highlighted specific challenges relating to the implementation of these principles. To our knowledge, no studies have attempted to identify practices in the rehabilitation setting.

The time dedicated to practicing graphomotor skills and handwriting at school has decreased over the last 30 years. In the USA in 1992, 85% of the school day was allocated to fine motor activities involving paper-and-pencil tasks [111]; in 2020, this time had decreased to between 18% and 37% of the school day (depending on the grades between kindergarten and fourth grade) [107]. However, the amount of practice time was found to vary greatly from one class to another (from 2 min to 1 h per day) in elementary school [108].

Most teachers propose a diverse range of task-specific activities for children to perform that are performed by the children in diverse conditions. Some practices appear to be common across teachers—for example, copying models of letters presented on the board, tracing letters in the air before writing them on the paper, performing pre-writing exercises or associating verbal cues with the tracing of letters, and allowing pupils to practice independently in a calligraphy book [108,109]. Depending on the country, the type of allograph taught may vary (for example cursive in France, script or mixed in Quebec) [109]. Some teachers who perform remediation also propose fine motor activities (finger games, modelling clay) or gross motor activities (activities to develop the body schema) to children without difficulties to facilitate handwriting skill development [110].

The quality and quantity of augmented feedback given (e.g., guiding the movement, modelling the shape of the letter by showing visually and describing verbally) vary greatly among teachers [108,109]. Teachers often provide knowledge of results through positive feedback when the letter is well formed, and they frequently encourage self-assessment by the pupil of their writing [108]. Teachers seem to consider their presence as non-essential for the pupil during the training and evaluation phases of the handwriting learning. Therefore, the knowledge of results delivered by teachers relates to the static characteristics of the writing and not the dynamics of the gesture [110].

Teaching handwriting in school is challenging for several reasons. First, only a minority (12%) of teachers consider themselves to be properly trained to teach handwriting [108]. Second, the group context makes it difficult to implement individualized instruction, to provide physical guidance [108], or to assess the dynamics of the handwriting gesture for each pupil. Third, despite the obvious benefits of collaboration between education and rehabilitation professionals, as well as the professionals’ willingness to collaborate around a child with difficulties, collaboration is hard to implement because of organizational determinants (i.e., facilitating planning, meetings, and information sharing) and systemic factors (the health and education systems are totally separate) [112].

### 4.5. Considerations

This second part of the review identified, from a research perspective, the main facilitating factors common to learning and remediation of handwriting: the duration of training, the individualization of objectives and learning paths, the importance of the learning context, the quantity and nature of feedback available to correct action execution and build motor programs as well as the benefits of individualized remediation. Further research remains necessary to determine the most effective interventions and modalities to facilitate handwriting learning and remediation for the children depending on their needs. In practice, the consideration of all these factors constitutes a real challenge. Studies on practices in the field of education have identified some common practices but also a great variability in the methods used by professionals, especially feedback provision. They highlight the lack of technical and human resources available to education professionals to characterize the needs of each child, to personalize learning paths, and to collaborate with rehabilitation professionals to improve the inclusion of children with special needs.

## 5. Handwriting Acquisition with Digital Learning Technologies

Digital technologies may facilitate pedagogical accessibility through consideration of the variability of learning processes and the creation of enabling learning situations for all [113]. From this point of view, digital technologies meet a number of challenges.

### 5.1. What Are Digital Learning Technologies

There is currently no general agreement on the definitions of the terms educational technology, assistive technology, and learning technology [114,115]. The term educational technology is used to denote the “processes, tools, equipment, devices and systems used to support and facilitate learning, teaching and assessment” [115]. “Assistive technology adds the notion of services, systems, processes and environmental modifications used by people with disability and/or older people to overcome the social, infrastructural and other barriers to (learning) independence, full participation in society and carrying out (learning) activities safely and easily” [115]. Learning technology encompasses both educational technologies and the assistive technologies used, for instance, to access educational technologies and/or learning materials, participate in learning activities, and/or overcome barriers to learning [115].

Digital technologies encompass various devices available on the market or specifically designed for handwriting. Virtual reality systems [116,117] and digital tablets have already been introduced into teaching and rehabilitation [24,25]. Sales of tablets have exceeded computer sales since 2015 [24] and will be specifically considered in this review. Other technologies, such as robotic devices [118,119], should also be considered as potential tools for handwriting practice.

Digital technologies can be divided into hardware and software. Hardware consists of physical devices and their components that can be seen and touched (e.g., laptops and tablets). Software consists of programs, routines, and symbolic languages that control the hardware; it does not have physical components (e.g., mobile apps and e-learning platforms [114].

Some educational software use AI. AI is capable of imitating certain functionalities of human intelligence, including features such as perception, learning, reasoning, problem solving, language interaction, and even producing creative work [120]. AI can be used to support learning through diverse potential applications [23]. Student-facing AI technologies have received the most attention from researchers, developers, educators, and policymakers and will be specifically considered in this section because they aim to provide the learner with access to augmented feedback and high-quality personalized learning and to facilitate new approaches to assessment [121].

Digital learning technologies might be used as (i) tutors (i.e., the technology provides lessons and practice adapted to the learners), (ii) teaching aids (i.e., the technology assists the teacher in decision making or provides models to assist instruction), (iii) tools for learning (i.e., technology provides feedback to the learner to facilitate the learning process), or (iv) tools to facilitate communication between stakeholders involved with children [122]. For learners with special needs, technology can compensate for impairments (e.g., by enlarging characters), or circumvent impairments (e.g., by reading an instruction by voice synthesis or by adapting the size of the characters according to the writer’s abilities). They can also support learning (e.g., by facilitating the recognition of written words for students with dyslexia). 

As such, AI could help address the various challenges faced by education and rehabilitation professionals to support the acquisition of handwriting for all children.

### 5.2. Evidence-Based Practices of Handwriting Learning with Digital Environments

Over the past 20 years, a variety of information communication technologies (ICT) and computer-assisted instruction interventions for learning to write have been introduced in the classroom. However, meta-analyses conducted specifically on handwriting have been unable to draw clear conclusions about the benefits of these devices [123,124]. The disparity in the results of the different studies likely relates to the wide variety of devices evaluated [123].

The meta-analysis by Wollscheid et al. (2015), which measured the impact of practicing with digital technologies (versus paper and pencil) on the quality and quantity of handwriting, showed that writing by hand always produced better results than typing [125]. However, only one study included in the meta-analysis [126] compared paper-pencil writing, typing, and tablet-stylus writing, limiting the conclusions regarding this question. A more recent meta-analysis by Ismail and Ghani (2021) highlighted the growing interest in the use of tablets to teach handwriting to children (7 of 13 studies focused on tablets) and the contrasting results of studies evaluating the benefits tablet use for writing [124]. Some research conducted over short periods of time (a few hours or few days) suggested that the difference in friction between writing on paper or a tablet disrupts the graphomotor strategies of children and adults [127,128], whereas other studies found positive results from learning to trace letters on a tablet with a stylus [129] or a finger [130]. Several hypotheses can be drawn from these studies, and further research is needed to identify why tablet learning degraded the handwriting of some children. The handling of a new tool (i.e., different degree of friction on the tablet and the narrow stylus) might be challenging, and associated changes in posture might disrupt the graphomotor strategies of novice writers [129,130,131]. These observations should lead professionals to choose equipment that considers the needs of young writers: the possibility to rest the wrist on the surface, use of a filter on the screen to imitate the paper friction, and an adapted stylus or finger guides to ensure an appropriate grasp. Another solution is to draw letters on the tablet with the fingers [129,130,131]. Furthermore, learning may be enhanced by the use of varied tools and supports; however, the learner must be given sufficient time to become experienced in the use of the tool [132].

## 6. Potentialities of Digital Learning Technologies for Handwriting

When considering digital technologies for handwriting learning, both the type of hardware and the software should be considered, since the richness and potentiality of a technological device are particularly related to the functionalities of the software. Research on digital technologies highlights their specific potential to improve learning, motivation, and cooperation. These technologies could facilitate understanding of handwriting difficulties by providing an enriched assessment. Those that include AI could also provide personalized handwriting learning for each child, facilitating inclusion. 

### 6.1. Evaluations Using Tablets

Digital technologies could accurately quantify aspects of therapy that cannot be measured clinically. Such information could increase understanding of remediation science in research and decision making in education and clinical practice. 

Digital handwriting evaluations using a stylus and tablet could objectify and automatize clinical handwriting tests by calculating static features that are similar to those obtained through paper evaluations. Size, space between words, density, and space between strokes have been shown to be relevant variables, and they indirectly highlight the features described by an evaluator such as readability and shape [133,134,135,136].

In addition to the static handwriting features, digital evaluations can assess kinematics and pen pressure and orientation (altitude and azimuth). The strokes can be considered as abnormal fluctuations in a velocity profile that can be analyzed with a signal-to-noise analysis [137], calculation of jerk (i.e., derivative of acceleration) [136,138] or the median of the power spectrum of speed frequencies [133]. Abnormal fluctuations of pressure can be identified by analyzing the power spectrum of speed frequencies of pressure change, of the median and variance of the pressure. Thus, beyond observable behavior, analysis of the gesture using a digital tablet can reveal unnecessary movements during writing, variability in the pressure exerted, and spatiotemporal inconsistency of the letters written [57].

Such evaluations have been shown to be valid compared to a classic handwriting evaluation using the HSPQ or the BHK [133,134,136,138,139,140,141]. Furthermore, digital evaluations can classify children with handwriting difficulties into specific clusters using pressure and pen orientation features [134]. Studies investigating handwriting in children with developmental coordination disorder [8,142,143,144] or ADHD [145,146,147,148] also identified differences in kinematics and pressures compared with typically developing children. Teachers and rehabilitation professionals could use these clusters to determine the specific needs of a child in terms of learning, but more research is needed to identify the real potential of digital evaluations to personalize learning paths. 

### 6.2. The Potential of AI for Handwriting Learning in the School Context: The Intuiscript Project

In 2017, a team of researchers, developers, and practitioners designed an AI-based solution as part of the Intuiscript project. This project aimed to produce an intelligent solution (Kaligo^®^, Learn & Go, Rennes, France) for a tablet equipped with a stylus to assist the teaching of graphomotor skills and to support learning by respecting the learning rhythm of each student. A digital workbook provides a dynamic model of the letter and extrinsic augmented feedback (i.e., explicit knowledge of the result) based on a fine analysis of the handwriting gesture. During activities, the part of the model to be copied (i.e., letter, bigram, trigram, or word) is highlighted (Figure 1). Before the child writes, the model is displayed dynamically to help them to write correctly. The feedback is based on the data collected by an automatic writing analysis processor that evaluates the size, order, direction, and shape of the strokes. These handwriting analyses are used to personalize the teaching scenarios. For example, the AI will elaborate specific exercises involving writing letters or segments of letters that have not been mastered for a student who repeatedly fails to write the letters of a word and to link them together. The teacher can use the application to personalize the learning path of each student and review the dynamic layout provided to each student and the feedback they receive. Children can also work autonomously by using the tablet workbook and following their personalized pathway. Thus, such a device can be used as a tutor and teaching aid as well as a learning tool. 

Several types of feedback, in the form of color markers, are produced and inform the child about the overall quality of their handwriting (as a gauge in the form of a horseshoe that is colored from red to green and is associated with a star if the writing is sufficiently consistent with the model).

A quasi-experimental study conducted in 2020 by Bonneton-Botté et al. (2020) evaluated the benefits of the application in a study involving 233 kindergarten students classified as low-, moderate-, and high-ability writers from 22 classes during a 12-week teacher-implemented program [132]. The control group, composed of 93 students, followed the usual school program, working exclusively on paper. The 138 students in the experimental group followed the usual program using paper but they also participated in additional workshops using tablets equipped with the Kaligo application (the overall duration of writing training was equivalent in both groups). The results showed that use of the Kaligo application led to larger improvements in handwriting for students with an initially moderate ability. For students with an initially low or high ability, the benefits were equivalent. The results of that study also emphasize that by giving students and teachers sufficient time to appropriate the tool, studies have a better chance of objectively measuring the benefits of digital technologies.

### 6.3. The Potential of Digital Technologies for Handwriting Remediation

Digital technologies can be used to propose modules that could benefit all children or subpopulations depending on their developmental level or their abilities. 

For example, some systems allow the creation of a module that associates (i) vocalization of instructions and letters, words, sentences, or text pronunciation and (ii) analysis of the oral production of the child’s voice during a reading activity and feedback. Such modules that are specifically dedicated to conceptual knowledge could help the child to acquire the main knowledge needed for handwriting acquisition (i.e., associate names, sounds, visual shapes, and gestures of letters; read pseudowords and obtain feedback on his/her reading).

Among digital technologies, commercial devices such as virtual reality (VR) or handwriting robots seem promising. These technologies stimulate active engagement, can provide auditory and/or visual feedback, and facilitate task repetition, factors that are key to improving motor performance [118,119,149,150,151]. VR may enhance the development of gross and fine motor skills in children with developmental disabilities [149,150,151]. Regarding handwriting remediation, VR could be used to improve the construction of motor representations of letters by involving the use of gross motor skills. Furthermore, VR can provide varied environments that cannot be provided otherwise, which may facilitate learning. 

The digitalization of handwriting practice can be used to diversify the types of feedback provided to the learner. For example, movement sonification can provide information about kinematics or pressure on the writing surface; acceleration peaks or significant pressure on the pen could produce a sound that indicates to the writer the need to regulate their movement [106]. This type of feedback could be added to handwriting software. 

Lastly, and in response to the need for collaboration between families, teachers, and rehabilitation professionals, as well as the involvement of parents in remediation, digitalization might facilitate communication between these stakeholders by removing organizational barriers that limit discussion [23,112,152]. By facilitating access to information, technologies may also provide an opportunity to empower parents and propose a family-centered approach to remediation. 

### 6.4. The Potential Offered by Big Data Collection and Artificial Intelligence

AI is a powerful tool that might open up the “black box” of learning [121]. Digitalization facilitates the collection of large volumes of multimodal data. Information can be gathered about children’s handwriting profiles as well as other determinants of handwriting ability, such as cognitive function, or the school and family environments, the learning pathway on the software with the activities they have completed, their performance on each activity, etc. [121]. The analysis of possible correlations and patterns across diverse data modalities is too complex for subjective analysis, and AI is well adapted for this purpose. First, AI techniques, such as deep learning, can be used to analyze this information and (i) help teachers and rehabilitation professionals to understand their students’ profiles and shape their future interventions appropriately, as well as (ii) enable the students to track their own progress [121]. Second, at a research level, the collection and analysis of multimodal and longitudinal data may facilitate the definition of developmental trajectories [153] that are useful in clinical practice to set adapted goals. It might also help to develop an integrated model of handwriting acquisition that takes into account the child’s motor and cognitive development. Finally, the collection of evaluation and curriculum data may help to formulate relevant and specific hypotheses on the therapeutic strategies that should be offered to children with handwriting disorders and that can be tested in studies designed to provide a high level of evidence [154]. 

### 6.5. Challenges for the Implementation of Digital Learning Technologies

Despite the captivating perspectives set out above, the translations of these innovations into practice might be limited. Although tablet apps may offer opportunities to support handwriting learning, lack of acceptance of new technologies by educational and rehabilitation professionals could be a barrier to the use of tablets as a tool for teaching or remediating handwriting [155,156,157]. The development and implementation of new technologies can only be successful if the emphasis is placed on user-centered design, implementation, and evaluation. Consideration of professionals’ views might drive successful future developments [21,158,159]. 

A new project, Kaligo + is being developed to further the Intuiscript project. The aim is to provide an enriched version of Kaligo specifically for use by children with handwriting disorders (whatever the cause) and all the actors involved in their care. The project has three main objectives: (i) to design a connected platform for collaboration between the different actors (i.e., therapists, teachers, child, parents, and researchers) to assist the diagnostic process, facilitate the continuity of the follow-up, and increase the efficiency of the intervention; (ii) to develop a remediation and training module for handwriting combining exercises on a tablet, on paper, and in virtual reality based on general principles but also recent research findings; (iii) to collect internationally big data to establish relationships between writers’ profiles and learning paths to move towards a greater personalization of handwriting learning in the future. To facilitate the translation of research results into practice as well as the implementation of this device, the whole application is being co-designed with future users (children, families, teachers, and rehabilitation professionals) and researchers.

## 7. Conclusions

Since prehistoric times, humans have desired to leave traces of their existence, first using handprints and then using tools as they began to develop. Although keyboards have become a relevant means that is prioritized by many adults for notetaking, handwriting has other functions in children. It contributes to the preparation of reading, writing, and spelling skills. Considering the importance of handwriting in childhood, digital learning environments should not be seen as an end point, but as a means to personalize learning paths to ensure success for all. Current school methods of remediation cannot be totally individualized; individualization could be achieved using digital environments with AI. In the context of inclusion, digital learning technologies might also facilitate collaboration between school and rehabilitation professionals. 

Digital environments offer great learning opportunities; however, several conditions must not be overlooked if these tools are to be useful, usable, and acceptable in school and rehabilitation settings. Such learning environments should not be used by practitioners until their impact and usefulness have been evaluated. Users should be systematically involved in designing the interface, the remediation and training exercises, and the functionalities to ensure the usability and acceptability of the tool. The development of a training policy and cooperative research including teachers, researchers, rehabilitation professionals, and users will help to avoid a Manichean perspective: being for or against digital learning environments for handwriting.

## Figures and Tables

**Figure 1 children-10-01096-f001:**
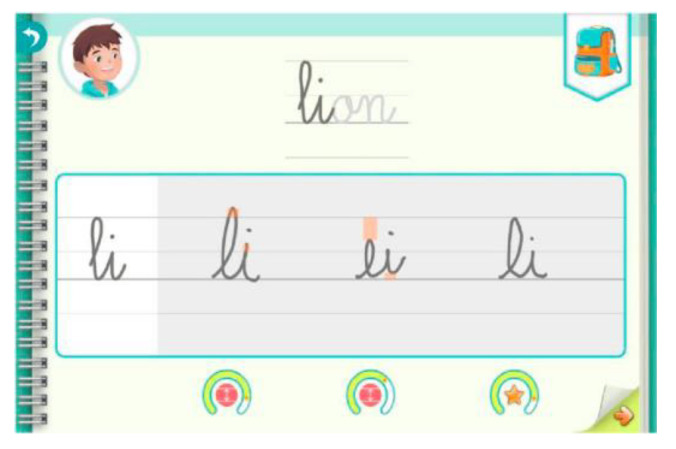
Three trials with feedback for the first syllable of the word “lion”.

## Data Availability

No new data were created or analyzed in this study. Data sharing is not applicable to this article.

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
