# Peer review of "Teaching and Rehabilitation of Handwriting for Children in the Digital Age: Issues and Challenges"

_children, 2023, doi:10.3390/children10071096_

Round 1

Reviewer 1 Report

REPORT

CHILDREN-“Teaching and rehabilitation of handwriting for children in the digital age: issues and challenges”.

In my judgment, the article is well-written and presents interesting information on the handwriting topic. However, even though it is a "Narrative Review", I request that the authors insert information about the method used.

Information such as the time period in which the searches were carried out; the databases; the descriptors; etc, are useful so that other researchers can, in the future, carry out other review studies.

Abstract:

pg 1 - lines 17 to 19

Introduction:

pg 2 - lines 51 to 53

The objectives of the study are slightly different, I suggest that the objectives in the Abstract and Introduction are the same.

Keywords:

Pg 1 - line 20

I suggest replacing the word handwriting with graphomotor or similar. This change is justified by the possibility of increasing the possibilities in searches carried out by other researchers.

Conclusions:

Pg 5 - lines 243 to 252

Pg 8 and 9 - lines 400 to 414

The manuscript has three "Conclusions". For better understanding of the reader, I suggest changing the title of the subsections on pages 5 and 8 to "Considerations".

Pg. 12

line 593 and 594

 change:

i/help Teachers and rehabilitation...

ii/enable the students to...

to:

(i) help Teachers and rehabilitation...

(ii) enable the students to...

Pg 13

Lines 615 to 620

Change:

i) to design a connected...

ii) to develop a remediation...

iii) to collect internationally...

To:

(i) to design a connected...

(ii) to develop a remediation...

(iii) to collect internationally...

REPORT

CHILDREN-“Teaching and rehabilitation of handwriting for children in the digital age: issues and challenges”.

In my judgment, the article is well-written and presents interesting information on the handwriting topic. However, even though it is a "Narrative Review", I request that the authors insert information about the method used.

Information such as the time period in which the searches were carried out; the databases; the descriptors; etc, are useful so that other researchers can, in the future, carry out other review studies.

Abstract:

pg 1 - lines 17 to 19

Introduction:

pg 2 - lines 51 to 53

The objectives of the study are slightly different, I suggest that the objectives in the Abstract and Introduction are the same.

Keywords:

Pg 1 - line 20

I suggest replacing the word handwriting with graphomotor or similar. This change is justified by the possibility of increasing the possibilities in searches carried out by other researchers.

Conclusions:

Pg 5 - lines 243 to 252

Pg 8 and 9 - lines 400 to 414

The manuscript has three "Conclusions". For better understanding of the reader, I suggest changing the title of the subsections on pages 5 and 8 to "Considerations".

Pg. 12

line 593 and 594

 change:

i/help Teachers and rehabilitation...

ii/enable the students to...

to:

(i) help Teachers and rehabilitation...

(ii) enable the students to...

Pg 13

Lines 615 to 620

Change:

i) to design a connected...

ii) to develop a remediation...

iii) to collect internationally...

To:

(i) to design a connected...

(ii) to develop a remediation...

(iii) to collect internationally...

Author Response

Dear Reviewer 1

We thank you for the time you have spent reading our paper and for your comments and suggestions which have helped to strengthen it. We have taken into account all the suggestions and hope that you will now find this revised version suitable for publication in Children. We asked our translator to check the English.

We have provided a point-by-point response to your comments below.

In my judgment, the article is well-written and presents interesting information on the handwriting topic.

However, even though it is a "Narrative Review", I request that the authors insert information about the method used.

Information such as the time period in which the searches were carried out; the databases; the descriptors; etc, are useful so that other researchers can, in the future, carry out other review studies.

Thank you for your comment, thank you for your suggestion which helped us to improve the manuscript. We have added a method section.

This narrative review was conducted according to the Scale for the Assessment of Narrative Review Articles (SANRA) criteria (Baetghe C 2019). It was specifically designed to consider and synthesize data from the educational and rehabilitation sciences regarding handwriting learning and the opportunities offered by new technologies for the acquisition of this skill for all children.

A literature search was performed between July 2022 and April 2023 by the 4 authors using the following terms: handwriting; dysgraphia; motricity; diagnosis; evaluation; analysis; intervention; rehabilitation; therapy; education; collaboration; technology; artificial intelligence. Relevant scientific studies were identified in Medline (Pubmed), Web of Science, google scholar IEEE and Scopus. A comprehensive search of the references of selected papers was conducted to identify further articles of interest. Special attention was paid to systematic reviews and meta-analyses. Articles published in French and English in peer reviewed journals were considered. We considered studies of new technologies published from 2010 to provide up-to-date data and trends in this field. We paid specific attention to devices that have already been introduced in teaching and rehabilitation programs.

We synthesized the evidence with regard to the goals of this narrative review. The authors met regularly to cross-check ideas and expertise to (i) identify and select key statements and (ii) organize the presentation of the data.”

Abstract:

pg 1 - lines 17 to 19

Introduction:

pg 2 - lines 51 to 53

The objectives of the study are slightly different, I suggest that the objectives in the Abstract and Introduction are the same.

Thank you for your comment. We modified the objectives in the abstract which are now the same as the objectives in the introduction. This narrative review aims to present and discuss the challenges of handwriting learning and the opportunities offered by new technologies involving AI for school and health professionals to successfully improve the handwriting skills of all children.”

Keywords:

Pg 1 - line 20

I suggest replacing the word handwriting with graphomotor or similar. This change is justified by the possibility of increasing the possibilities in searches carried out by other researchers.

Thank you for your suggestion. We added graphomotor learning as a sixth keyword to increase its visibility in searches carried out by other researchers.

Conclusions:

Pg 5 - lines 243 to 252

Pg 8 and 9 - lines 400 to 414

The manuscript has three "Conclusions". For better understanding of the reader, I suggest changing the title of the subsections on pages 5 and 8 to "Considerations".

 Thank you for your comment. We changed that.

Pg. 12 line 593 and 594

 change: i/help Teachers and rehabilitation... ii/enable the students to...

to: (i) help Teachers and rehabilitation... (ii) enable the students to...

Pg 13 Lines 615 to 620

Change: i) to design a connected... ii) to develop a remediation...iii) to collect internationally...

 To: (i) to design a connected... (ii) to develop a remediation... (iii) to collect internationally...

 Thank you for your comment. We changed that.

Reviewer 2 Report

This paper reviews work on the difficulties of learning handwriting and proposes methods and techniques that may be used to ameliorate the problem in some children. The paper is excellent and it is going to be both theoretically and practically useful. 

I have a minor suggestion for improvement. The discussion on the developmental and individual differences possible causes of difficulties in learning to write is rather limited. For instance, I would like to see a more thorough discussion of the developmental changes that take place from about 4 to 7 years that relate to handwriting, such as symbolic and representational awareness, interlinking representations and action, controlling attention focus, etc. The development of these processes is obviously related to learning to write and also to the ability to use the technology proposed in the paper. I suggest that this literature is searched and more adequately discussed in the paper. 

Author Response

Dear reviewer 2

We thank you for the time you have spent reading our paper and for your comments and suggestions which have helped to strengthen it. Many thanks for your encouraging feedback and for the time you have devoted to reviewing this work We have taken into account your suggestions and hope that you will now find this revised version suitable for publication in Children.

This paper reviews work on the difficulties of learning handwriting and proposes methods and techniques that may be used to ameliorate the problem in some children. The paper is excellent and it is going to be both theoretically and practically useful. 

I have a minor suggestion for improvement. The discussion on the developmental and individual differences possible causes of difficulties in learning to write is rather limited. For instance, I would like to see a more thorough discussion of the developmental changes that take place from about 4 to 7 years that relate to handwriting, such as symbolic and representational awareness, interlinking representations and action, controlling attention focus, etc. The development of these processes is obviously related to learning to write and also to the ability to use the technology proposed in the paper. I suggest that this literature is searched and more adequately discussed in the paper. 

Your proposal is particularly interesting. Although our original intention was to deal with handwriting from a motor point of view, the cognitive dimension, and in particular the role of conceptual knowledge, received too little attention in the first version of the manuscript. We have added a paragraph to clarify a few developmental benchmarks relating to the acquisition of conceptual knowledge (cf. 3.2. Developmental benchmarks for the acquisition of handwriting)

 “From a cognitive point of view, conceptual knowledge of written language could go a long way towards explaining inter-individual variability. Children between the ages of 3 and 5 years acquire both universal and specific knowledge However, according to Puranik and Lonigan (2011), children first need to acquire universal knowledge about the written language system (i.e., when they pretend to write, young children produce linear, segmented strokes composed of simple shapes) and then specific knowledge linked to the linguistic system (i.e., directionality, symbol shape associated with the name and sound of letters) [4]. Between the ages of 5 and 7 years, children develop the ability to write letters from memory and fluently [44,45], which is essential for orthographic and compositional skills. On the graphomotor level, writing consists of transcribing the letters that make up words, and a delay in the conceptual knowledge of linguistic units might contribute to explain the variability observed on the gestural level. These developmental sequences can guide understanding of how gaps are created between pupils during the first years of school.”

As a result, the contribution perspectives relating to new technologies have also been enriched (cf.5.3. The potential of digital technologies for handwriting remediation).

“Digital technologies encompass various devices available on the market or specifically designed for handwriting. Virtual reality systems [113,114] and digital tablets have already been introduced into teaching and rehabilitation [115,116]. Sales of tablets have exceeded computer sales since 2015 [115] and will be specifically considered in this review. Other technologies, such as robotic devices [117,118], should also be considered as potential tools for handwriting practice.”

Reviewer 3 Report

For a good and actualized narrative review, you should include new theoretical perspectives, as differential learning and non-linear pedagogy.

Likewise, you should include other technological (AI) proposals, e.g., among many others: Guneysu Ozgur, A., Özgür, A., Asselborn, T., Johal, W., Yadollahi, E., Bruno, B., ... & Dillenbourg, P. (2020). Iterative design and evaluation of a tangible robot-assisted handwriting activity for special education. Frontiers in Robotics and AI, 29.

Author Response

Dear reviewer 3,

We would like to thank you for the thoughtful comments and the time dedicated to the evaluation and revision of our manuscript. We have taken into account all the suggestions and hope that you will now find this revised version suitable for publication in Children.

We have provided a point-by-point response to your comments below.

For a good and actualized narrative review, you should include new theoretical perspectives, as differential learning and non-linear pedagogy.

We would like to thank you for this very interesting suggestion. Our methodology and the keywords used (which have been clarified in a new paragraph following the suggestions of reviewer 1) did not lead us to identify the non-linear pedagogy approach. We find this approach interesting in that it is in line with most of the recommendations we made during our review. To introduce this theoretical field, we have inserted a reference to the non-linear theories of motor learning and a reference to the field of non-linear pedagogy.

“Our literature search did not yield articles about the non-linear dynamics of motor learning theory; however, this theory might be considered as a complementary approach [32].”

And

“Those principles are in accordance with the principles of motor learning theories and the principles of non-linear pedagogy [87], and emphasize the relevance of providing personalized handwriting instruction.”

Likewise, you should include other technological (AI) proposals, e.g., among many others: Guneysu Ozgur, A., Özgür, A., Asselborn, T., Johal, W., Yadollahi, E., Bruno, B., & Dillenbourg, P. (2020). Iterative design and evaluation of a tangible robot-assisted handwriting activity for special education. Frontiers in Robotics and AI, 29.

Thank you for your comment. We mainly focused on devices already available on the market. We now expose our strategies of research in a method section and open the possibility to investigate other technologies.

“2. Methods

This narrative review was conducted according to the Scale for the Assessment of Narrative Review Articles (SANRA) criteria (Baetghe C 2019). It was specifically designed to consider and synthesize data from the educational and rehabilitation sciences regarding handwriting learning and the opportunities offered by new technologies for the acquisition of this skill for all children.

A literature search was performed between July 2022 and April 2023 by the 4 authors using the following terms: handwriting; dysgraphia; motricity; diagnosis; evaluation; analysis; intervention; rehabilitation; therapy; education; collaboration; technology; artificial intelligence. Relevant scientific studies were identified in Medline (Pubmed), Web of Science, google scholar IEEE and Scopus. A comprehensive search of the references of selected papers was conducted to identify further articles of interest. Special attention was paid to systematic reviews and meta-analyses. Articles published in French and English in peer reviewed journals were considered. We considered studies of new technologies published from 2010 to provide up-to-date data and trends in this field. We paid specific attention to devices that have already been introduced in teaching and rehabilitation programs.

We synthesized the evidence with regard to the goals of this narrative review. The authors met regularly to cross-check ideas and expertise to (i) identify and select key statements and (ii) organize the presentation of the data.”

And

 “Indeed, new technologies encompass various devices available on the market or specifically designed for handwriting. Virtual reality systems (Dolgunsöz et al. 2018, Lamb et Etopio, 2019) and digital tablets have already been introduced into teaching and rehabilitation (Kavanagh et al., 2017). Sales of tablets have exceeded computer sales since 2015 (Hassler et al., 2016) and will be specifically considered in this review. Other technologies, such as robotic devices (Chandra et al. 2019, Ozgur et al., 2020), should be considered as potential tools for handwriting practice.”

Round 2

Reviewer 3 Report

no further comments